# APP Knock-In Mice Produce E22P-Aβ Exhibiting an Alzheimer’s Disease-like Phenotype with Dysregulation of Hypoxia-Inducible Factor Expression

**DOI:** 10.3390/ijms232113259

**Published:** 2022-10-31

**Authors:** Takahito Maki, Masahito Sawahata, Ichiro Akutsu, Shohei Amaike, Genki Hiramatsu, Daisuke Uta, Naotaka Izuo, Takahiko Shimizu, Kazuhiro Irie, Toshiaki Kume

**Affiliations:** 1Department of Applied Pharmacology, Graduate School of Medicine and Pharmaceutical Sciences, University of Toyama, Sugitani, Toyama 930-0194, Japan; 2Department of Pharmaceutical Therapy and Neuropharmacology, Graduate School of Medical and Pharmaceutical Sciences, University of Toyama, Sugitani, Toyama 930-0194, Japan; 3Aging Stress Response Research Project Team, National Center for Geriatrics and Gerontology, Obu 474-8511, Japan; 4Division of Food Science and Biotechnology, Graduate School of Agriculture, Kyoto University Kitashirakawa-Oiwake-Cho, Kyoto 606-8502, Japan

**Keywords:** Alzheimer’s disease, amyloid β, knock-in mice, toxic conformer, oligomer, tau phosphorylation, cognitive function, synaptic plasticity, glial activation, hypoxia-induced factor

## Abstract

Alzheimer’s disease (AD) is a progressive neurodegenerative disorder that requires further pathological elucidation to establish effective treatment strategies. We previously showed that amyloid β (Aβ) toxic conformer with a turn at positions 22–23 is essential for forming highly toxic oligomers. In the present study, we evaluated phenotypic changes with aging in AD model *App^NL-P-F/NL-P-F^* (NL-P-F) mice with Swedish mutation (NL), Iberian mutation (F), and mutation (P) overproducing E22P-Aβ, a mimic of toxic conformer utilizing the knock-in technique. Furthermore, the role of the toxic conformer in AD pathology was investigated. NL-P-F mice produced soluble toxic conformers from an early age. They showed impaired synaptic plasticity, glial cell activation, and cognitive decline, followed by the accumulation of Aβ plaques and tau hyperphosphorylation. In addition, the protein expression of hypoxia-inducible factor (HIF)-1α was increased, and gene expression of HIF-3α was decreased in NL-P-F mice. HIF dysregulation due to the production of soluble toxic conformers may be involved in AD pathology in NL-P-F mice. This study could reveal the role of a highly toxic Aβ on AD pathogenesis, thereby contributing to the development of a novel therapeutic strategy targeting the toxic conformer.

## 1. Introduction

Alzheimer’s disease (AD) is a progressive neurodegenerative disease that leads to cognitive decline and accounts for the most significant proportion of dementia patients [1]. The insufficient understanding of AD makes it challenging to develop a treatment strategy, and further elucidation of AD pathology is an urgent issue. On the other hand, considerable evidence has previously been accumulated on AD pathology, and the amyloid cascade hypothesis is widely supported [2,3,4]. According to this hypothesis, AD pathology is characterized by the accumulation of amyloid β-protein (Aβ), followed by neurofibrillary tangles (NFT) due to tau protein hyperphosphorylation, and finally, brain atrophy results in cognitive decline. Other important features of AD include impaired synaptic plasticity [5,6,7]. Long-term potentiation (LTP) in the hippocampus is considered the synaptic basis of learning and memory formation [8,9,10]. The LTP suppression observed in AD patients and AD model mice correlated with cognitive decline [11,12,13]. Previously, we reported that monomers with a turn at positions 22–23 of Aβ have high aggregation ability and potent cytotoxicity in vitro. We named Aβ with these structures as toxic conformers [14,15,16,17]. To investigate the importance of toxic conformers in vivo, we generated *APP^NL-P-F/NL-P-F^* mice (NL-P-F mice) with the Aβ precursor protein (APP) gene modified using the knock-in method [18]. NL-P-F mice showed cognitive dysfunction at six months and Aβ deposition in the brain at eight months [19]. It has been suggested that this mouse model is a helpful tool for assessing oligomer toxicity because it exhibits an oligomer-driven phenotype by producing toxic conformers of Aβ, which quickly form oligomers (Appendix A). However, how toxic conformers contribute to the progression of AD pathology has not been studied in detail. This study investigated age-dependent phenotypic changes related to AD pathology in NL-P-F mice.

In addition, neuroinflammation caused by glial cell activation plays an essential role in AD [20,21,22,23,24]. Activated microglia and reactive astrocytes surround Aβ plaques (gliosis) and induce neuroinflammation [25,26,27,28]. Thus, it was suggested that glial cells act as regulators of inflammation in the brain. It has been previously demonstrated that the modulation of the hypoxia-inducible factor (HIF) pathway of glial cells is a potential therapeutic target for AD [29,30]. Aβ plaque accumulation and Aβ oligomer administration increase HIF-1α expression in microglia, suggesting the involvement of Aβ production and HIF-1 pathway activation [31,32]. HIF-3α forms heterodimers with HIF-1α and HIF-1β but has no transcriptional activity and represses downstream gene expression in the HIF-1 pathway [33,34]. However, few reports have been mentioned on the involvement of HIF-3α in AD. Thus, we also investigated the potential of the HIF pathway as a therapeutic target in NL-P-F mice.

## 2. Results

### 2.1. The Number of Toxic Conformers Increases with Age in the Brain of NL-P-F Mice

We first investigated the number of toxic conformers from three to twelve months. The toxic conformer levels were measured by sandwich ELISA with 24B3 antibody, which specifically recognizes the toxic conformer structure, turn at positions 22–23, and 82E1 antibody, which recognizes the N-terminus of Aβ [35]. The TBS-soluble toxic conformers were significantly higher in NL-P-F mice (NLPF) from three to twelve months than in wild-type (WT) (Figure 1A). The TBS-insoluble fraction showed no difference between wild-type and NL-P-F mice for three to six months. The TBS-insoluble toxic conformers were significantly higher in NL-P-F mice from nine to twelve months (Figure 1B).

### 2.2. Aβ Plaques Accumulate in the Hippocampus and Cortex of NL-P-F Mice from Nine Months Onward, with Tau Hyperphosphorylation in the Hippocampus at Twelve Months

Next, we examined whether the specific histological changes found in AD patients occur in NL-P-F mice. The significant pathological signs of AD include Aβ accumulation followed by excessive tau phosphorylation and neurodegeneration. Aβ deposition was observed in the hippocampal cornu ammonis 1 (CA1) and dentate gyrus (DG), and the cerebral cortex (CRT) by nine months in NL-P-F mice (Figure 2A). In contrast, no Aβ deposition was observed in wild-type mice by twelve months. Quantification of the Aβ deposition area revealed a significant increase in Aβ deposition in all observed areas of NL-P-F mice after nine months (Figure 2B). To examine the levels of phosphorylated tau, we performed immunohistochemical staining with an anti-paired helical filaments-1 (PHF-1) antibody (phosphorylated at Ser396/Ser404). Hyperphosphorylated tau was detected in hippocampal sections at twelve months (Figure 2C). Well-stained areas in the hippocampal sections were the pyramidal cell layer of hippocampal CA3 and CA1, and the subgranular zone, and the polymorphic layer of the DG (Appendix A). Quantification of the p-tau immunoreactive area was significantly higher in NL-P-F mice after twelve months (Figure 2D,E). There was no significant change between wild-type and NL-P-F mice at nine months (Appendix A). We next examined the extent of neuronal death at twelve months by immunostaining for hippocampal neuronal nuclei (NeuN) (Figure 2F,G). The number of NeuN-positive cells in hippocampal CA1 and CA3 was similar by twelve months (Figure 2H).

### 2.3. Cognitive Function Declines after 6 Months of Age in NL-P-F Mice

In a previous study, NL-P-F mice showed cognitive dysfunction at six months of age [19], but it has not yet been clarified when cognitive dysfunction begins to appear. To assess cognitive function, we performed the novel object recognition (NOR) test, which measures object recognition, and the Y-maze test, which measures spatial cognition (Figure 3A). The discrimination index of the NOR test and the alternation of the Y-maze test also indicate long-term and working memory formation, respectively. In both tests, cognitive function was comparable at three months, whereas after six months, cognitive function was significantly reduced in NL-P-F mice (Figure 3B,C). The total number of touching objects in the NOR test and the arm entries in the Y-maze test were almost equivalent between wild-type and NL-P-F mice from three to twelve months (Appendix A).

### 2.4. Synaptic Plasticity Is Reduced in the Hippocampal CA1 Region of NL-P-F Mice after Three Months

We prepared acute hippocampal slices of NL-P-F mice and assessed paired-pulse ratio (PPR) and LTP indicators of synaptic plasticity from three to twelve months. PPR measurements showed no significant differences between age-matched wild-type and NL-P-F mice from three to twelve months (Figure 4A). Quantification of LTP by the area under the curve (AUC) of fEPSPs slope after high-frequency stimulation (HFS) showed that LTP was suppressed in hippocampal slices of NL-P-F mice after three months (Figure 4B,C). At three and twelve months, the input–output curves showed a slightly downward trend in NL-P-F mice but were almost the same as in wild-type mice (Appendix A).

### 2.5. Glial Cell Activation Occurred for Three Months Even in the Absence of Aβ Deposition, and Gliosis Was Observed after Aβ Plaque Deposition

Next, we examined the extent of neuroinflammation in NL-P-F mice from three to twelve months of age by triple immunohistochemical staining of microglia, astrocytes, and Aβ. Notably, the fluorescent signal of ionized calcium-binding adapter molecule 1 (Iba1) and glial fibrillary acidic protein (GFAP) was increased without Aβ deposition in NL-P-F mice by three months (Figure 5A). In NL-P-F mice, microglia accumulated, and astrocytes were activated at the sites of Aβ deposition (Figure 5B). We then quantified the fluorescent area and fluorescence intensity of these images. In hippocampal CA1 and DG, Iba1-positive area and GFAP fluorescence signal intensity were increased in NL-P-F mice, especially by three months before Aβ deposition (Figure 5C,D).

### 2.6. Dysregulation of Hypoxia-Inducible Factor (HIF) Expression in Hippocampal Tissue of NL-P-F Mice at Six Months of Age or Later

Previous reports showed that the HIF pathway of glial cells could be a potential therapeutic target for AD [29,30]. We, therefore, examined the protein and mRNA expression levels of HIF subtypes from three to twelve months of age. The protein expression of HIF-1α increased at six months (Figure 6A). In contrast, the gene expression of HIF-3α was decreased in NL-P-F mice from six to nine months of age (Figure 6B). Gene expression HIF-1α and HIF-1β expression were comparable in wild-type and NL-P-F mice (Appendix A).

## 3. Discussion

A previous study using NL-P-F mice showed that Aβ oligomers were formed after six months of age, resulting in increased toxic conformer levels and deposition of Aβ plaques after eight months [19]. Here, NL-P-F mice had significantly increased toxic conformer levels by three months in the TBS-soluble fraction and nine months in the TBS-insoluble fraction. NL-P-F mice had Aβ deposition from six to nine months of age and increased tau phosphorylation after twelve months. NL-P-F mice also showed deficits in cognitive function after six months of age. Synaptic plasticity was impaired in the hippocampus of NL-P-F mice by three months of age. In the hippocampus of the NL-P-F mice, astrocytic and microglial activation occurred by three months. These findings suggest that NL-P-F mice produce soluble toxic conformers that cause glial cell activation and reduced synaptic plasticity in the hippocampus via oligomer formation. NL-P-F mice were also valuable as a model of AD caused by Aβ production, which reproducibly causes cognitive impairment after six months.

### 3.1. The Production of Toxic Conformers Promotes Tau Hyperphosphorylation but Not Neuronal Loss

The toxic conformer levels in NL-P-F mice increased with aging, whereas their levels were very low in wild-type mice throughout the assessment months in this study. 24B3, developed by immunization of a toxic conformer surrogate E22P-Aβ9-35 in mice, was helpful for AD diagnosis using human cerebrospinal fluid (CSF) [35]. It is suggested that the 24B3 antibody recognizes the toxic conformation of wild-type Aβ aggregate forms, such as oligomers [36]. In this study, wild-type mice from three to twelve months showed little or no production of Aβ oligomers. Various reports suggest that the presence of Aβ contributes to tau hyperphosphorylation and neuronal loss [37,38,39]. In the present study, phosphorylated tau was significantly increased in the hippocampus of NL-P-F mice after twelve months (Figure 2E). In addition, the ratio of toxic conformer to total Aβ_1-42_ in cerebrospinal fluid was significantly higher in AD [40].

Furthermore, toxic conformers and phosphorylated tau levels are increased in 3 × Tg insulin-deficient AD model mice, and toxic conformers are co-localized with tau oligomer [41]. The results of the above clinical studies and the present study suggest that toxic conformers can affect tau hyperphosphorylation. Tau hyperphosphorylation is known to cause neurofibrillary changes and, ultimately, neuronal degeneration. Therefore, we investigated whether NL-P-F mice producing E22P-Aβ caused neuronal death in the hippocampus. However, by twelve months, neuronal death did not occur in NL-P-F mice (Figure 2H). Few AD model mice generated based on Aβ pathology show an early neuronal loss, and significant neuronal loss was observed in transgenic mice crossed with many AD-related gene mutations (5 × FAD mice) [42].

On the other hand, P301S Tg mice overexpressing humanized tau develop filamentous tau lesions after six months of age. By nine to twelve months, marked neuronal loss occurs along with atrophy of the hippocampus and entorhinal cortex [43]. Furthermore, APP/Tau double transgenic mice expressing human mutant APP and human mutant tau show neuronal loss in the entorhinal cortex after nine months of age compared with APP alone transgenic mice, tau alone transgenic mice, and wild-type mice [44]. These reports suggest that the progression of tau pathology is essential for neuronal loss that closely reproduces clinical AD pathology. Further progression of tau pathology would be necessary for neuronal loss in NL-P-F mice. It is necessary to verify whether further aging and progression of tau pathology will cause a neuronal loss in future studies. Here, we observed that the production of toxic conformers promoted tau hyperphosphorylation in the hippocampus of NL-P-F mice.

### 3.2. Toxic Conformers Cause Impairment of Synaptic Plasticity and Cognitive Function

In this study, the NL-P-F mice showed impaired long-term memory formation in the NOR test from six to twelve months of age (Figure 3B). We found that working memory formation in the Y-maze test was impaired from six to twelve months of age (Figure 3C). Aβ deposition by six months was minimal, and Aβ plaques were significantly higher during nine to twelve months of age (Figure 2B). These results suggest that the cognitive decline was due to soluble Aβ oligomers rather than the accumulation of Aβ plaques. This early cognitive decline is suggested to be an oligomer-driven phenotypic change characteristic of NL-P-F mice. In support of this notion, reduced LTP (Figure 4) and glial activation (Figure 5) were shown prior to Aβ plaques in NL-P-F mice. Intraventricularly injected Aβ oligomers impair memory in mice and rats and reduce LTP induction in the hippocampus [45,46]. Reflux administration of Aβ oligomers to acute hippocampus slices in wild-type mice also causes deficits in synaptic plasticity in the hippocampus [47,48]. Thus, Aβ impairs synaptic plasticity and causes cognitive decline. The present study showed suppression of LTP induction in hippocampal slices at three months of age, preceding the cognitive decline (Figure 4C). In addition, reflux administration of E22P-Aβ_1-42_ peptide inhibits LTP induction [49]. Given the high levels of soluble toxic conformers in NL-P-F mice at three months of age, it is likely that the impairment of synaptic plasticity is partially due to soluble toxic conformers present in hippocampal tissue.

In contrast, PPR was not significantly different between wild-type and NL-P-F mice (Figure 4A). LTP is involved in postsynaptic and PPR in presynaptic plasticity [50,51,52]. Thus, it is suggested that postsynaptic plasticity is impaired in the hippocampal CA1 region of NL-P-F mice at least after three months of age. However, LTP induction in NL-P-F mice tended to decrease but did not show a significant decline at nine months of age. (Figure 4C). LTP in NL-P-F mice tended to increase from nine to twelve months of age compared to an increase from three to six months (Figure 4C), but cognitive function continued to decline (Figure 3). In AD model mice Tg2576, LTP induction increases during older ages due to the disinhibition of excitatory synapses by a decrease in inhibitory interneurons [53]. In addition, LTP is increased in APP with “Osaka mutation (E693Δ)” homozygous KI mice compared to wild type due to suppression of GABAergic synapses [54]. As shown above, cognitive function declines with aging even when LTP induction increases. The disinhibition of inhibitory synapses may be one of the reasons why the induction of LTP in NL-P-F mice did not continuously decline with aging. Further studies must determine whether excitatory synaptic disinhibition occurs in NL-P-F mice during aging.

### 3.3. Toxic Conformers Induce Neuroinflammation through Glial Cell Activation

Interestingly, astrocytes and microglia were activated three months before forming Aβ plaque deposition (Figure 5C,D). Previous reports showed that resting microglia increase in the hippocampal CA1 region prior to Aβ plaque formation and extracellular Aβ accumulation in 3 × Tg mice [55]. In the hippocampus of hAPP-J20 mice, the number of astrocytes and microglia increased before Aβ deposition [56]. In the cortex, Iba1-positive microglia clustered at the Aβ plaque deposition site, and GFAP fluorescence intensity increased surrounding the Aβ plaques (Figure 5B). In contrast, in the hippocampus, the activation of astrocytes was observed from three to twelve months of age, independent of the area of Aβ deposition (Appendix A). In AD model mice, GFAP fluorescence intensity of astrocytes shows different changes depending on the brain region [57]. Astrocytes play the broadest homeostatic function in the central nervous system and are diversely involved with AD disease [58,59]. One reason astrocytes in the NL-P-F mice are activated differently in each brain region may be that astrocyte activation plays different roles in different brain tissue. Microglial activation was observed in the NL-P-F mice from three to twelve months of age compared with wild-type mice detected by increased Iba1 fluorescence area in the hippocampus and cortex, regardless of the site of Aβ plaque deposition (Figure 5C, Appendix A). Inflammatory cytokines are released via the NF-kB pathway when microglia are exposed to Aβ fibril [60]. In response to the accumulation of Aβ oligomers in neurons early in the pathology, the number of microglia increases, and cell bodies enlarge in the hippocampus before Aβ plaque deposition in the AD model mice [61]. The present study also suggests that neuroinflammation was induced by the activation of microglia with toxic conformers. Toxic conformation-restricted Aβ_1-42_ with an intramolecular disulfide bond showed increased uptake into THP-1 macrophage-like cells and significantly higher cytotoxicity compared to wild-type Aβ_1-42_ and E22P-Aβ_1-42_ at low concentrations [62]. Intramolecular disulfide bonds strongly induce toxic conformations of Aβ and stabilize the oligomer formation. This suggests that toxic conformers form soluble oligomers that activate microglia, leading to oligomer-driven progression of AD pathology.

### 3.4. Dysregulation of HIF-Related Molecules Contributes to the Progression of AD-Related Pathology in NL-P-F Mice

In order to determine the mechanisms of neuroinflammation with toxic conformers, we focused on the changes in the HIF pathway. The protein levels of HIF-1α were significantly increased in the hippocampus of the NL-P-F mice at six months (Figure 6A). HIF-1α is degraded by the proteasomal pathway under normoxic conditions, whereas under hypoxic conditions, the protein is stabilized and translocated into the nucleus, leading to the expression of downstream target genes. In addition, the gene expression levels of HIF-3α were reduced in the hippocampus of the NL-P-F mice from six to nine months (Figure 6B). HIF-3α is a downstream target gene of HIF-1α, and its expression was increased in response to hypoxia [63]. In addition, HIF-3α acts as negative feedback to regulate gene expression in HIF-1α downstream pathways by competitively inhibiting the transcriptional activity of other HIFs [33]. However, the protein expression of HIF-1α increases in NL-P-F mice at six months. The negative feedback of HIF downstream gene expression may be disrupted by decreased gene expression of HIF-3α. Previously, it was reported that upregulation of HIF-1α increased β-site APP-cleaving enzyme1 (BACE1) gene expression and Aβ production in neurons [64,65]. Additionally, HIF-1α expression induces inflammation via NF-kB expression [66]. The dysregulation of HIF pathway expression in NL-P-F mice could induce increased Aβ production and neuroinflammation, which may contribute to the onset of AD pathology. Microglia, the leading cause of inflammation in the brain, are suggested to enhance migration to Aβ plaques and phagocytosis through activation of the mTOR-HIF-1α pathway via Trem2 [67]. Trem2-mediated microglial activation may be partially responsible for underlying the increased HIF-1α protein expression in NL-P-F mice. However, glial cells were activated by three months, preceding the dysregulation of HIF expression at six months. Pathways other than the HIF pathway may be involved in microglial activation. In addition, this study did not examine the involvement of the HIF pathway in each cell type. The role of the HIF pathway in microglial activation by toxic conformers needs to be examined in detail by future studies. To summarize, toxic conformers activate the HIF-1 pathway from three to six months of age, increasing soluble Aβ production. However, they may suppress Aβ metabolic degradation function from nine to twelve months of age due to reduced activation of the HIF-1 pathway.

### 3.5. Limitation

This study has potential limitations. In this study, NL-P-F mice exhibited AD-related phenotypic changes, such as Aβ accumulation and tau hyperphosphorylation, but no neuronal loss. These phenotypic changes were preceded by cognitive decline. In clinical AD patients, Aβ accumulation, tau hyperphosphorylation, and brain atrophy with neurodegeneration ultimately lead to cognitive decline. Therefore, a gap exists in the progression of AD between the NL-P-F mouse and the clinical pathology. It is necessary to be careful in comparing this mouse model with the clinical pathology. In this study, we demonstrated that regulation of HIF-1α and HIF-3α expression may be a potential target for AD therapy. On the other hand, previous studies have shown inconsistent expression levels of HIF-1α in other AD model mice [68,69,70]. Protein expression levels of HIF-1α may vary depending on the AD model mice and the experimental design, such as timing and region of tissue sampling. In addition, there are few reports on the relationship between HIF-3α expression levels and AD pathology. Further studies on the relationship between decreased HIF-3α gene expression levels and phenotypic changes in AD are required. Some limitations regarding HIF expression should be noted in this study. The present study only shows in vivo phenotypic changes over time. Future studies, therefore, should examine whether modulating the expression levels of HIF-1α or HIF-3α by genetic or pharmacological approaches can ameliorate the AD-related phenotypic changes in NL-P-F mice. It will also be necessary to determine what molecular mechanisms, directly or indirectly, lead the toxic conformers to disrupt HIF signaling.

## 4. Materials and Methods

### 4.1. Animals

As described previously, NL-P-F mice models were generated with a C57BL/6 genetic background [19]. NL-P-F mice were bred by crossing heterozygous mice, and the genotype was determined using primers (Thermo Fisher Scientific, Carlsbad, CA, USA) with the following sequence: 5′-AAGG-TAGCTTGGCTGTCCTTT-3′ (forward primer) and 5′-TTTTTCTCCTAAGTGGCCCCG-3′ (reverse primer). Homozygous NL-P-F mice and their wild-type littermates were used in this study. Animals were maintained in a 25 ± 1 °C room, with 55 ± 2% relative humidity, under a 12 h light/dark cycle (7 a.m.–7 p.m.), with ad libitum access to water and food. We followed the guidelines of the Japanese Pharmacological Society regarding animal experiments and received appropriate education and training for animal experiments. Additionally, all the experiments were conducted following the ethical guidelines of the University of Toyama Animal Experiment Committee and with its approval (A2021PHA-14). Behavioral tests and brain tissue sampling were conducted at three, six, nine, and twelve months, respectively.

### 4.2. Tissue Preparation

The tissue preparation was completed as previously described [71] with minor modifications. Mice were anesthetized with a mixture of three anesthetic agents: 0.75 mg/kg medetomidine hydrochloride (Nippon Zenyaku Kogyo, Koriyama, Japan), 4.0 mg/kg midazolam (Sandoz K.K., Tokyo, Japan), and 5.0 mg/kg butorphanol tartrate (Meiji Seika Pharma, Tokyo, Japan). After anesthesia, mice were perfused transcardially with phosphate-buffered saline (PBS, 1.76 mM KH_2_PO_4_, 2.7 mM KCl, 10 mM Na_2_HPO_4_, 137 mM NaCl, pH 7.4), and their brains were collected. After the brain was removed, the hemisphere was sliced coronally at 6 mm from the olfactory bulbs using a brain slicer (MK-MC-01, Muromachi Kikai, Tokyo, Japan) and divided into anterior tissue for ELISA and hippocampus for Western blot and real-time reverse transcription (RT)-PCR. Each tissue was flash-frozen in liquid nitrogen and stored at −80 °C for biochemical analysis. Another hemisphere was immersed and fixed in 4% paraformaldehyde (PFA) solution (Wako, Osaka, Japan) for 2 h at 4 °C and used for immunohistochemical staining.

### 4.3. ELISA

ELISA was performed according to a previous study with minor modifications [72]. Frozen brain tissue was crushed rapidly with the SK mill (SK-200, Token, Chiba, Japan) and mixed with tris-buffered saline (TBS, 50 mM Tris-HCl, 138 mM NaCl, and 2.7 mM KCl) containing 1% protease inhibitor cocktail set III dimethyl sulfoxide solution (Wako) and 1 mM phenylmethylsulphonyl fluoride (PMSF) on ice. After centrifugation (20,000× *g*, 4 °C, 5 min), the supernatant was used as the TBS-soluble fraction. The pellet was dissolved in TBS containing 6 M guanidine-HCl (Nacalai Tesque, Kyoto, Japan) and incubated on ice for 10 min. The lysate was centrifuged (20,000× *g*, 4 °C, 30 min), and the supernatant was used as the TBS-insoluble fractions. According to the instruction manual, the ELISA kit (Cat#27709, Immuno-Biological Laboratories, Gunma, Japan) was used to determine the concentration of the toxic conformers.

### 4.4. Immunohistochemistry (IHC)

Immunohistochemical staining techniques were slightly modified from previous studies [73]. PFA-fixed tissues were dehydrated in 30% sucrose at 4 °C until they sank. After dehydration, the tissues were embedded in an optimal cutting temperature (OCT) compound (Sakura Finetek, Tokyo, Japan) and frozen at −80 °C. Frozen tissues were sliced coronally to prepare 30 μm thick brain sections containing the cortex and hippocampus behind the bregma (Bregma −2.0 to −2.4 mm), a cryostat (Leica CM 3050S, Leica Biosystems, Nussloch, Germany). Brain sections were washed three times for 5 min with PBS and stored in an antifreeze solution (30% ethylene glycol, 30% glycerol, 40% PBS, 0.05% Sodium azide) at −20 °C. For staining, brain sections were washed with PBST (0.3% Triton X-100 in PBS) and blocked with a blocking buffer (1% donkey serum and 1% bovine serum albumin (BSA), 0.05% sodium azide in PBST) for 60 min at room temperature. Each primary antibody (Table 1) was diluted in a blocking buffer and reacted with brain sections overnight at 4 °C. After washing with PBST; brain sections were incubated with a secondary antibody (Table 2) diluted in PBST for 120 min at room temperature. Brain sections were washed three times and then mounted in a mounting medium with or without DAPI (VECTASHIELD Mounting Medium, VECTOR Laboratories, Burlingame, CA, USA).

The fumigation fixation method invented by Dr. Miyasaka (Doshisha University, Kyoto, Japan) was used (Japanese patent application No. 2019-099443) to detect tau phosphorylation. Brain sections were steamed with 4% PFA steam. After washing for 5 min, brain sections were soaked in TBS containing 10% sodium dodecyl sulfate (SDS) for 5 min. The sections were washed for 5 min as a pretreatment before IHC.

For image analysis, fluorescence images were captured using an all-in-one fluorescence microscope (BZ-X800, Keyence, Osaka, Japan) and a confocal laser microscope (Zeiss LSM 900 with airy scan, Carl Zeiss, Oberkochen, Germany). The images were captured in the hippocampus and cortex (Appendix A) and quantified using 1–3 adjacent images from 3–4 brain sections per mouse in each group. Quantification of fluorescence images was performed according to previous reports [74,75,76]. GFAP and Iba-1 are the markers of astrocytes and microglia, respectively. Microglial activation leads to the enlargement of the cell body; therefore, the fluorescence area of Iba1 was measured. The fluorescence intensity of GFAP was measured to evaluate the reactive astrocyte. The number of NeuN-positive cells was counted in the area of the pyramidal cell layer within 600 μm along the hippocampal CA1 region and 500 μm along the hippocampal CA3 region to evaluate the neuronal loss. All fluorescence images were quantified using ImageJ/Fiji (National Institutes of Health, Bethesda, MD, USA).

### 4.5. Behavioral Tests

NOR test was carried out following the method mentioned in a previous study [19]. Before the NOR test, the mice were placed in a box (30 cm × 30 cm × 30 cm) for 10 min for five consecutive days to allow habituation to the experimental environment. In the acquisition phase, mice were placed facing two similar objects (familiar objects) for 10 min for three consecutive days. The following day in the test phase, mice were placed facing a familiar and novel object for 10 min on the fourth day. The number of times the mice touched the objects with their nose was counted to obtain the following discrimination index: Discrimination index = (number of times the mice touched the novel object/number of times the mice touched the novel object and familiar object) × 100.

The Y-maze test was carried out, followed by the method in the previous study [77]. Each arm was 41.5 cm long, 10 cm high, 4 cm wide at the bottom, and 10 cm wide at the top (YM-03M, Muromachi Kikai). In the Y-maze test, mice were placed in a maze of three arms and allowed to explore for 8 min freely. The total number of times the mice entered the arms and the number of alternations were counted to obtain the following alternation response rate. Alternation response rate = (number of alternations/total number of entries − 2) × 100.

### 4.6. Hippocampal Slice Preparation and Electrophysiological Recordings

The method of hippocampal slice preparation was the same as followed in a previous study [78,79]. After anesthesia, the mice were sacrificed by decapitation, and the entire brain was removed. The brain was immediately soaked for 3 min in ice-cold modified artificial cerebrospinal fluid (mACSF, 222.1 mM sucrose, 27 mM NaHCO_3_, 1.4 mM NaH_2_PO_4_, 2.5 mM KCl, 0.5 mM ascorbic acid, 1 mM CaCl_2_, and 7 mM MsgSO_4_). Appropriate portions of the brain were trimmed and placed on the ice-cold stage of a vibrating tissue slicer (VT-1000S, Leica Biosystems), and the brain tissue was cut into horizontal sections to prepare slices. The thickness of each tissue section was 300 μm. Hippocampal slices were incubated at 32 °C for 1 h in oxygenated (95% O_2_, 5% CO_2_) artificial cerebrospinal fluid (ACSF, 124 mM NaCl, 3 mM KCl, 26 mM NaHCO_3_, 2 mM CaCl_2_, 1 mM MgSO_4_, 1.25 mM KH_2_PO_4_, and 10 mM d-glucose). A slice was placed in the center of a multielectrode dish (MED probe, Alpha MED Science, Osaka, Japan). This device has an array of 64 planar microelectrodes, each having a size of 50 μm × 50 μm, arranged in an 8 × 8 pattern with an interpolar spacing of 150 μm [80]. The surface of the MED probe was coated with 0.1% polyethyleneimine (Sigma) in 25 mM borate buffer (pH 8.4) overnight at 4 °C to improve cellular adhesion.

The measurement of LTP was partially modified from the method used in a previous study [81,82]. During electrophysiological recordings, the slices were placed on the MED probe in a small CO_2_ incubator at 32 °C. Oxygenated and fresh ACSFs were infused at 1.5 mL/min. Evoked fEPSPs at all 64 sites were recorded with a multichannel recording system (MED64 system, Alpha MED Science) at a 20 kHz sampling rate and simultaneously filtered through a 100 Hz bandpass filter. One of the planar electrodes was used as a stimulating cathode. One of the electrodes in the Schaffer collateral/commissural fibers was selected as a stimulating electrode to collect typical responses in CA1.

In contrast, another in the stratum radiatum (dendritic region) was selected as a recording electrode. fEPSPs were recorded in response to test stimuli at excitatory synapses consisting of Schaffer collateral inputs from hippocampal CA3 and pyramidal cell dendrites of CA1. In each experiment, maximal fEPSPs were first determined by gradually increasing stimulus intensity until the saturation level was reached. Stimulus intensity was decreased to evoke a test response of approximately 30–50% of the maximal signal amplitude. PPR was determined by calculating the ratio of the average amplitude of the second response to the first. Inter-pulse intervals of the paired-pulse stimulation (PPS) were 25, 50, 100, and 200 ms. During baseline recordings, a single test pulse was delivered every 60 s for 40 min. After stable baseline recording, LTP was elicited by HFS protocols, delivering stimuli for 4 s with an interstimulus interval at 200 ms and each stimulus consisting of four pulses at 100 Hz. Data were collected for 60 min after inducing LTP. The fEPSP slope was normalized to the average value of the first 20 min of baseline (normalized fEPSP slope (%)). The AUC of the normalized fEPSP slope after LTP induction was calculated and evaluated as the magnitude of LTP induction (baseline = 0) for statistical analysis. The time course of the electrophysiological recording is shown in Appendix A. PPS was conducted to examine PPR 15 min before LTP measurement.

### 4.7. Western Blot

The Western blot experiment was performed as previously described [83] with minor modifications. The frozen tissue was crushed rapidly with the SK mill (Token) and dissolved in radioimmunoprecipitation assay (RIPA) buffer (50 mM Tris-HCl (pH 7.6), 50-mM NaCl, 1% Nonidet P40 substitute, 0.5% sodium deoxycholate, and 0.1% SDS) with 1% protease inhibitor cocktail (Wako), and 1 mM PMSF. After centrifugation (20,000× *g*, 4 °C, 5 min), the supernatant was collected, and the protein concentration was determined by a bicinchoninic acid protein assay kit (Takara). The protein samples were separated by sodium dodecyl sulfate–polyacrylamide gel electrophoresis (SDS-PAGE) and transferred to a polyvinylidene difluoride (PVDF) membrane (Bio-Rad). After blocking with 5% skimmed milk in TBS with Tween 20 (TBST, 10 mM Tris-HCl (pH7.5), 100 mM NaCl, and 0.1% Tween 20) at room temperature for 60 min, rabbit anti-HIF-1α (1:1000; NB100-479, Novus Biologicals, USA) or mouse anti-βactin (1:100,000; A5316, Sigma) antibody diluted in can obtain signal solution 1 (Toyobo, Osaka, Japan) was added and the membrane was incubated overnight at 4 °C. The membrane was incubated with horseradish peroxidase-conjugated anti-rabbit or anti-mouse IgG (1:5000; GE Healthcare, Buckinghamshire, UK) antibody diluted in can obtain signal solution 2 (Toyobo) for 90 min at room temperature. The signals were visualized by ECL detection reagents (GE Healthcare) using a LAS-4000 system (GE Healthcare), and the signal intensity was quantified using ImageJ/Fiji (National Institutes of Health). The HIF-1α protein level was normalized to β-actin as an internal reference in each sample.

### 4.8. Real-Time RT-PCR

The PCR experiment was performed as previously described [84] with minor modifications. The brain tissue was crushed with the SK mill (Token) and mixed with TRIzol (Thermo Fisher Scientific) for RNA extraction. According to the procedure manual, total RNA was purified by NeucleoSpin RNA (Takara, Shiga, Japan). The reverse transcription reaction was performed at 15 °C for 37 min, followed by inactivation of the enzyme at 85 °C for 5 s. The cDNA obtained was stored at −80 °C before PCR experiments. Real-time PCR reactions were performed using a Stratagene Mx3000P multiplex quantitative PCR system (Agilent Technologies Ltd., Santa Clara, CA, USA). The cycling conditions for cDNA amplification were 30 s at 95 °C and then 40 cycles of 5 s at 95 °C, 30 s at 55 °C, and 60 s at 72 °C. The mRNA expression was analyzed using Mx Pro QPCR software version 4.10 (Agilent Technologies). The primer sequences used for PCR were based on previous studies (Table 3) [63]. Data were analyzed by the 2^–ΔΔCT^ method and shown as a fold change in age-matched WT mice (% of WT) according to a previous study [85,86].

### 4.9. Statistical Analysis

The data were expressed as the mean ± standard error of measurement (SEM). Two-way analyses of variance (ANOVA) followed by Tukey’s post hoc test were used to analyze the statistical significance of differences among three or more groups of 4–15 mice. Comparisons between the two groups were tested for significant differences using the student’s *t*-test or Mann–Whitney rank sum test. All statistical analyses were performed using SigmaPlot 11.0 (Systat Software, San Jose, CA, USA) software packages. Probability values (*p*-value) < 0.05 were considered to be statistically significant. Detailed information on statistical analysis is shown (Appendix A).

## 5. Conclusions

This study performed phenotypic AD analyses in NL-P-F mice that physiologically produce toxic conformers. The following conclusions were acquired regarding the possible implications of toxic conformers in vivo for the first time: soluble toxic conformers were produced within three months, deposition of Aβ plaques increased significantly after nine months, and tau hyperphosphorylation occurred at twelve months of age. These findings indicate that toxic conformers cause tau hyperphosphorylation. These model mice are, therefore, expected to be useful to investigate the relationship between Aβ pathology and tau pathology. Cognitive impairment of long-term and working memory was observed after six months. This cognitive decline was partially due to impaired synaptic plasticity observed after three months. Glial cells were activated by three months, and soluble toxic conformers caused neuroinflammation even before the Aβ plaques were deposited. These results indicate that the NL-P-F mice are AD model mice characterized by oligomer-driven AD-related phenotypic changes caused by toxic conformers. These model mice are, therefore, expected to be useful to investigate the effect of oligomer formation on AD-related phenotypic changes. HIF-1α protein expression increased at six months, and HIF-3α gene expression decreased from six to nine months. After six months, this dysregulation of HIF-related molecules was suggested to be the molecular basis for AD pathogenesis in NL-P-F mice. This is the first study to demonstrate that toxic conformers contribute to the onset of AD-related pathology in vivo and that dysregulation of HIF-related molecular expression caused by decreased gene expression of HIF-3α and increased protein expression of HIF-1α is involved in the formation of early AD pathology that results in Aβ accumulation. Further clarification of the molecular mechanism by which the toxic conformers regulate the expression of HIF-related molecules in AD pathology could contribute to developing anti-Aβ therapies targeting the toxic conformers.

## Figures and Tables

**Figure 1 ijms-23-13259-f001:**
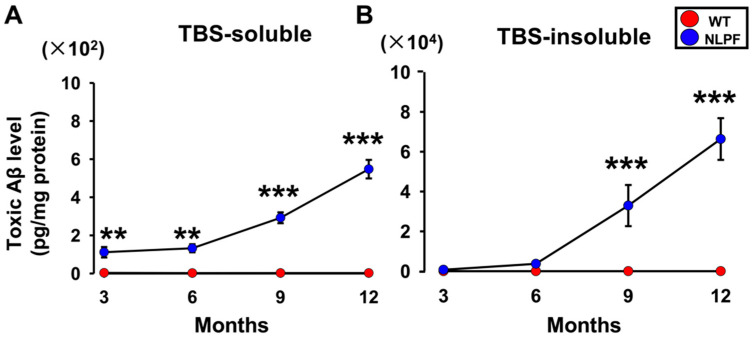
The toxic conformer accumulation in the brain of *App^NL-P-F/NL-P-F^* mice for three to twelve months. The levels of the toxic conformer of Aβ in (**A**) TBS-soluble and (**B**) TBS-insoluble fractions in the brains of *App^NL-P-F/NL-P-F^* mice from three to twelve months. (**A**) WT (*n* = 4–5), NLPF (*n* = 5). (**B**) WT (*n* = 4–5), NLPF (*n* = 4–5). The values indicate the mean ± SEM. ** *p* < 0.01, *** *p* < 0.001, compared with age-matched WT.

**Figure 2 ijms-23-13259-f002:**
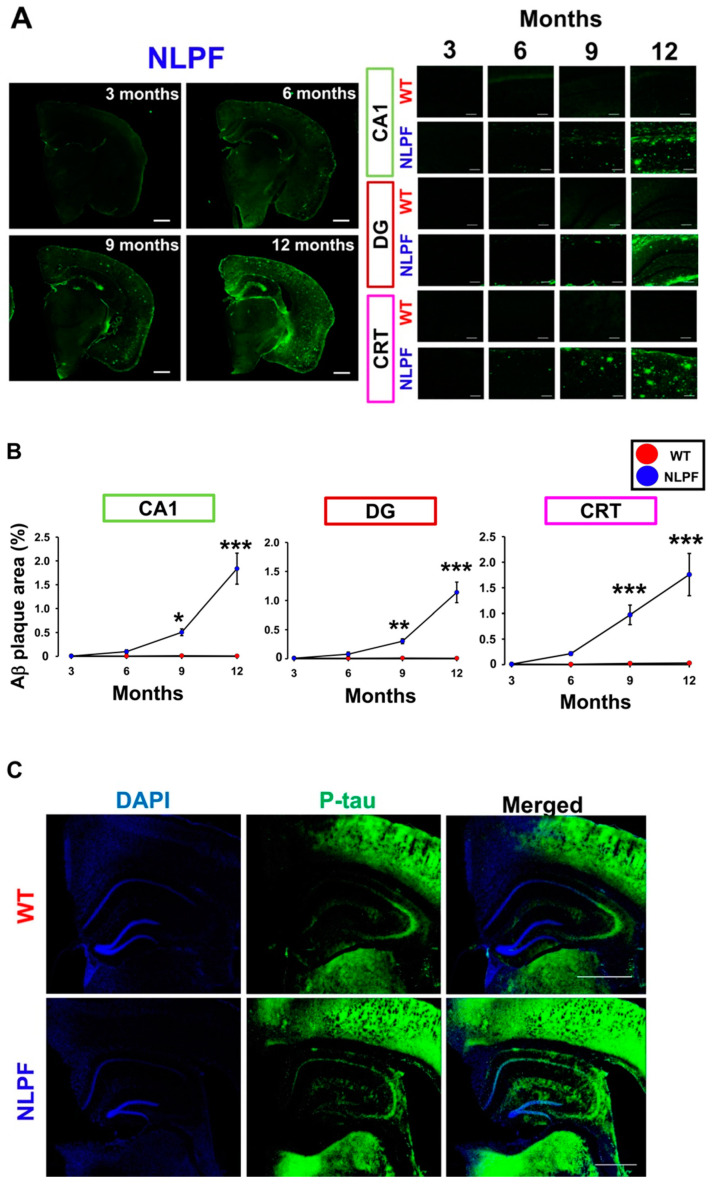
Alzheimer’s disease (AD)-related histological changes in the brain of *APP^NL-P-F/NL-P-F^* mice. (**A**) The Aβ plaque deposition was detected by 82E1 in the hippocampal CA1, DG, and cortex of *APP^NL-P-F/NL-P-F^* mice from three to twelve months. The scale bar indicates 1 mm in representative coronal sections and 100 μm in each region). (**B**) The Aβ plaque area in the hippocampal CA1, DG, and the cortex of *APP^NL-P-F/NL-P-F^* mice from three to twelve months. WT (*n* = 6) and NLPF (*n* = 7–9). (**C**,**D**) Representative (**C**) fluorescent and (**D**) confocal images of tau phosphorylation were detected by PHF-1 (Ser396/Ser404) in the hippocampal CA1, CA3, and DG after twelve months. The scale bar indicates (**C**) 1 mm (50 μm in enlarged images) and (**D**) 50 μm (25 μm in enlarged images of CA1 area). (**E**) After twelve months, the phosphorylated tau-positive area (% of total area) in the hippocampal CA1, CA3, and DG. WT and NLPF (*n* = 6). (**F**,**G**) After twelve months, representative (**F**) fluorescent and (**G**) confocal images of NeuN in the hippocampal CA1 and CA3. The scale bar indicates (**F**) 250 μm (50 μm in enlarged images) and (**G**) 50 μm. (**H**) After twelve months, the number of NeuN-positive neurons in the hippocampal CA1 and CA3. WT (*n* = 17 slices/6 mice), NLPF (*n* = 13 slices/6 mice). The values indicate the mean ± SEM. * *p* < 0.05, ** *p* < 0.01, *** *p* < 0.001, compared with age-matched WT.

**Figure 3 ijms-23-13259-f003:**
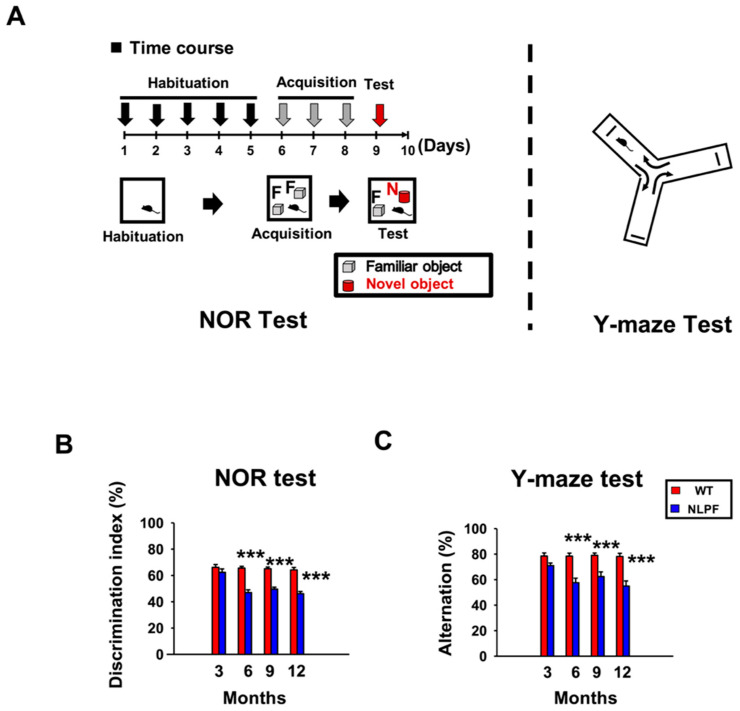
Cognitive function of *APP^NL-P-F/NL-P-F^* mice from three to twelve. (**A**) Time course and diagram of the behavioral test. (**B**) Discrimination index of novel object recognition (NOR) test. (**C**) Spontaneous alternation (%) of Y-maze test. WT (*n* = 10–15), NLPF (*n* = 10–15). The values indicate the mean ± SEM. *** *p* < 0.001, compared with age-matched WT.

**Figure 4 ijms-23-13259-f004:**
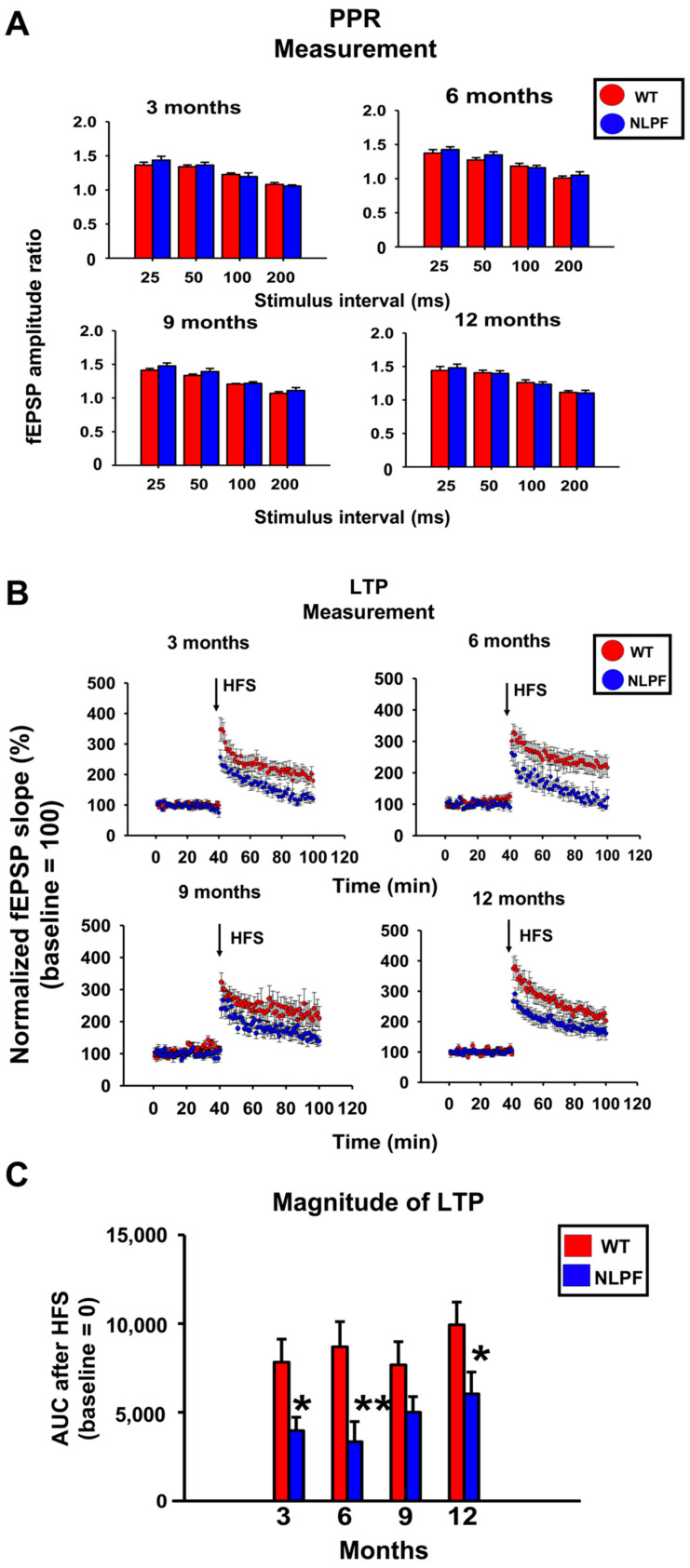
From three to twelve months, extracellular recordings in the CA1 of hippocampal slices from *APP^NL-P-F/NL-P-F^* mice. (**A**) Paired-pulse ratio (PPR) measurement. WT (*n* = 8–11 slices/4–5 mice), NLPF (*n* = 7–9 slices/4–5 mice). (**B**) Time course of fEPSP slope of Schaffer collateral-evoked synaptic responses in LTP measurement. (**C**) Post-HFS area under the curve (AUC) over the baseline of the evoked response in LTP measurement. WT (*n* = 8–10 slices/4–5 mice), NLPF (*n* = 7–10 slices/4–5 mice). The values indicate the mean ± SEM. * *p* < 0.05, ** *p* < 0.01, compared with age-matched WT.

**Figure 5 ijms-23-13259-f005:**
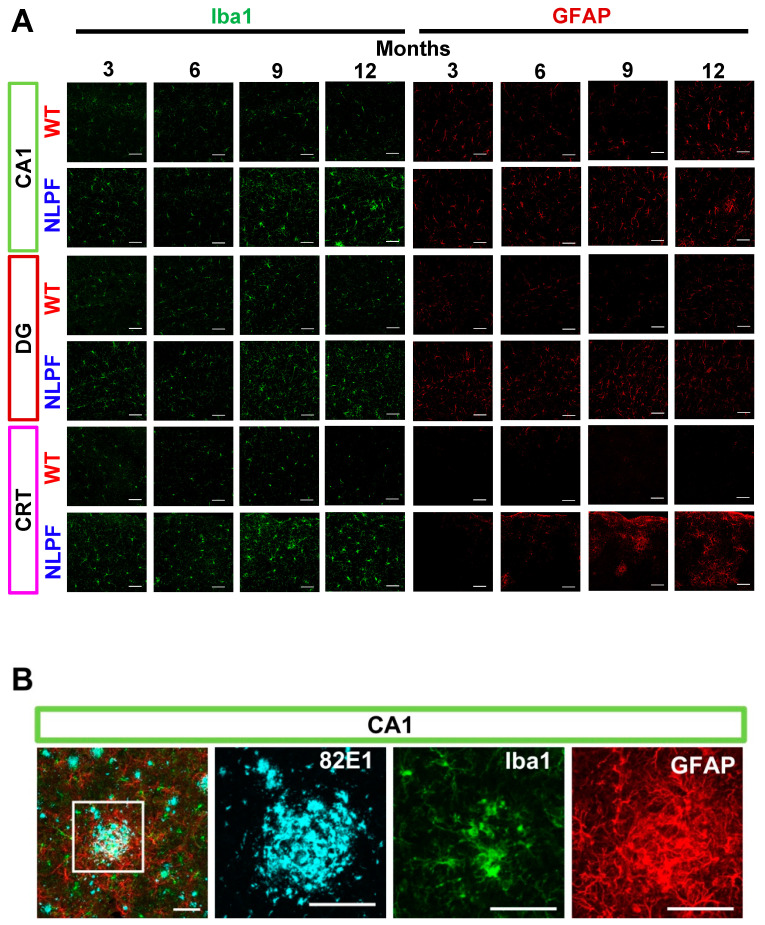
Neuroinflammation in the brain of *APP^NL-P-F/NL-P-F^* mice. (**A**) Iba1 and GFAP detected the glial cell activation in the hippocampal CA1, DG, and the cortex of *APP*^NL-P-F/NL-P-F^ mice from three to twelve months of age. The scale bar indicates 50 μm. (**B**) Representative images of gliosis in the cortex of *APP^NL-P-F/NL-P-F^* mice at twelve months of age. The scale bar indicates 50 μm. (**C**) Iba1 fluorescence area (% of total area) and (**D**) GFAP fluorescence intensity (per pixel) in the hippocampal CA1, DG, and the cortex of *APP^NL-P-F/NL-P-F^* mice from three to twelve months of age. WT (*n* = 5–6), NLPF (*n* = 7–9). Values indicate the mean ± SEM. * *p* < 0.05, ** *p* < 0.01, *** *p* < 0.001, compared with age-matched WT.

**Figure 6 ijms-23-13259-f006:**
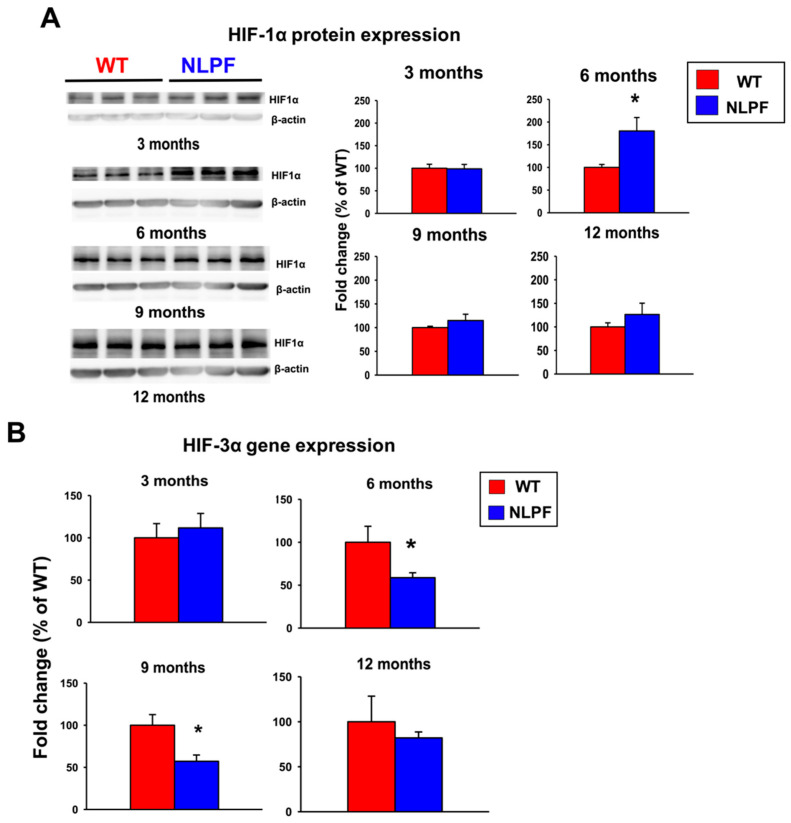
Relative expression of HIF in the hippocampal tissue of *APP^NL-P-F/NL-P-F^* mice from three to twelve months of age. (**A**,**B**) Fold change in (**A**) HIF-1α protein and (**B**) HIF-3α mRNA gene expression levels. (**A**) WT and NLPF (*n* = 4). (**B**) WT (*n* = 5–6), NLPF (*n* = 6). Values indicate the mean ± SEM. * *p* < 0.05 compared with age-matched WT.

**Table 1 ijms-23-13259-t001:** Primary antibodies used for immunohistochemistry (IHC).

Primary Antibody			
Product Name	Dilution Ratio	Supplier	Cat. No.
Rabbit anti-Iba1 antibody	1:1000	Wako, Osaka, Japan	019-19741
Mouse Anti-Human Amyloid β antibody (82E1)	1:1000	Immuno-Biological Laboratories, Gunma, Japan	10323
Goat Anti-GFAP antibody	1:1000	Sigma, St. Louis, MO, USA.	SAB2500462
Mouse anti-NeuN antibody, clone A60	1:300	EMD Millipore, Inc., Billerica, MA USA	MAB377
Mouse anti-PHF-1 antibody	1:250	Distributed by Dr. Peter Davies	

**Table 2 ijms-23-13259-t002:** Secondary antibodies used for immunohistochemistry (IHC).

Secondary Antibody			
Product Name	Dilution Ratio	Supplier	Cat. No.
Alexa Fluor 488 donkey anti-mouse IgG (H+L)	1:1000	Molecular Probs, Inc., Eugene, OR, USA	A-21202
Alexa Fluor 594 donkey anti-rabbit IgG (H+L)	1:1000	Invitrogen, Inc., Carlsbad, CA, USA	A-21207
Alexa Fluor 405 donkey anti-goat IgG (H+L)	1:1000	Abcam PLC, Cambridge, UK	Ab175664

**Table 3 ijms-23-13259-t003:** Primers used for RT-PCR.

Gene Name	Forward Primer Reverse Primer	Length (bp)	Gene Bank Accession No.
GAPDH	5′-CCAAGGTCATCCATGACAAC-3′ 5′-TTACTCCTTGGAGGCCATGT-3′	422	AB017801
HIF-1α	5′-AAGAAACCGCCTATGACGTG-3′ 5′-CCACCTCTTTTTGCAAGCAT-3′	301	AF057308
HIF-1β	5′-GCAGGATCAGAACACAGCAA-3′ 5′-CCTGGGTAAGGTTGGAGTGA-3′	301	U61184
HIF-3α	5′-AGAGAACGGAGTGGTGCTGT-3′ 5′-ATCAGCCGGAAGAGGACTTT-3′	301	NM022528

## Data Availability

Data are available on request due to restrictions, e.g., privacy or ethical. The data presented in this study are available on request from the corresponding author.

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
