# Peer review of "APP Knock-In Mice Produce E22P-Aβ Exhibiting an Alzheimer’s Disease-like Phenotype with Dysregulation of Hypoxia-Inducible Factor Expression"

_ijms, 2022, doi:10.3390/ijms232113259_

Round 1

Reviewer 1 Report

The study design is well done and the results and method sections are well written. Gap is not well mentioned in the introduction, and at the end of the discussion, it is better to add research limitations and suggestions for future research.

It is suggested that Table No. 4 be moved in the appendix of the paper.

Author Response

Thank you very much for your comments and suggestions, we revised the manuscript based on your input.

Point 1.
The study design is well done and the results and method sections are well written. Gap is not well mentioned in the introduction, and at the end of the discussion, it is better to add research limitations and suggestions for future research.

Response 1. 
Thank you for your comment. We added limitation and future study in this study at the end of the discussion. (pages 13/22 lines 383–403)

Point 2. 
It is suggested that Table No. 4 be moved in the appendix of the paper.

Response 2. 
Thank you for your comment. We added Table 4 as Table S1 to the supplementary data. (pages 18/22 lines 606)

Reviewer 2 Report

I suggest adjust the min max of the Fluorescence images properly so that the features are visible

Author Response

Thank you very much for your comments and suggestions, we revised the manuscript based on your input.

Point 1.
I suggest adjust the min max of the Fluorescence images properly so that the features are visible

Response 1. 
Thank you for your comment. We adjusted the contrast intensity and size of the immunostained images (Figure 2, 5) to make the difference between WT and NLPF more apparent. 

Reviewer 3 Report

In this manuscript, Maki et al further characterized a mouse model of AD (APP KI mice expressing E22P-Aβ), which was previously generated by the same group. The authors report that from early on (3 months), levels of soluble toxic Aβ conformers and inflammation were increased while LTP in CA1 was impaired, Aβ plaques appeared later (after 6 months) and p-Tau could be observed more later (after 12 months). Cognitive deficits stared to appear before the deposition of Aβ. In terms of the mechanism, the authors believe that the inflammation-induced changes of HIF expression were involved.

New models are certainly welcomed in the research field of AD. However, it is not clear what the advantage of this specific AD mouse model is in the context of the currently existing AD models including APP KI mice and mice overexpressing mutant human APP.

The resolution of all the fluorescent images in figures 2 and 5 was not high enough. On the other hand, widefield images containing the cortex and subregions of the hippocampus may be better than the images showing separated areas to show the distribution of the Aβ deposition (Fig. 2A). In Fig. 5A, what’s the point to merge the signals of Iba1 and GFAP which label different types of glial cells? In addition, GFAP is generally not expressed in the middle layers of the cortex in adult mice. The authors may want to use other markers such as S100β or ALDH1L1 to label astrocytes in the cortex.

When we talk about neuroinflammation, the morphology instead of the number of microglia/ astrocytes may be more informative.

The authors reported that “the protein expression of HIF-1α increased after six months (Figure 6A) (page 10/26, line186-187). However, Fig. 6A shows that there is no difference in the protein expression of HIF-1α between WT and NLPF at 9 and 12 months. Fig. 6B shows the HIF-3α gene expression of WT and NLPF, what about the protein expression of HIF-3α?

It seems that the change of both HIF-1α and HIF-3α was not age-dependent in NLPF mice. Why is that? How do you want to put this kind of variation into the context of the pathology and cognitive deficits of the NLPF mice?

The author may want to check the grammar throughout the manuscript including the title. For instance, the text description regarding the mouse ages is confusing.

Author Response

Thank you very much for your comments and suggestions, we revised the manuscript based on your input.

Point 1. 
New models are certainly welcomed in the research field of AD. However, it is not clear what the advantage of this specific AD mouse model is in the context of the currently existing AD models including APP KI mice and mice overexpressing mutant human APP.

Response 1. 
Thank you for your comment. We have found in a previous study that NL-P-F mice are AD model mice with oligomer-mediated cognitive decline that precedes Aβ deposition (Izuo, N., 2019). In this study, we found that NL-P-F mice exhibit cognitive decline after synaptic plasticity impairment and neuroinflammation. This model mice are thus an AD model mice characterized by the production of oligomer-driven AD-related phenotypic changes due to toxic conformers. We consider that this model mice will be useful to investigate the effect of oligomer formation on AD-related phenotypic changes. In addition, this mouse model exhibits tau hyperphosphorylation in the hippocampus after 12 months of age. Among AD model mice based on the Aβ hypothesis, few models induce tau hyperphosphorylation. Therefore, this mouse model is expected to be a useful tool to investigate the relationship between Aβ pathology and tau pathology. We added the descriptions about the characteristics of this mouse model in Conclusion. (pages 18/22 lines 614–616, pages 18/22 lines 619–623)

Point 2. 
The resolution of all the fluorescent images in figures 2 and 5 was not high enough. On the other hand, widefield images containing the cortex and subregions of the hippocampus may be better than the images showing separated areas to show the distribution of the Aβ deposition (Fig. 2A). In Fig. 5A, what’s the point to merge the signals of Iba1 and GFAP which label different types of glial cells? In addition, GFAP is generally not expressed in the middle layers of the cortex in adult mice. The authors may want to use other markers such as S100β or ALDH1L1 to label astrocytes in the cortex. 

Response 2. 
Thank you for your comments. We have improved the resolution of many of the images in Figures 2 and 5, making them easier to see by enlarging the images and enhancing the contrast. For Aβ deposition, we added a representative stained images of a coronal section of the brain hemisphere of NL-P-F mice, including the hippocampus. We have summarized the immunostained images of Iba1 and GFAP respectively, except for the merge image of Iba1 and GFAP in Figure 5A. Thank you also for your suggestion about astrocyte labeling markers. GFAP has been used to assess astrocyte labeling and activation in the cortex in various previous studies using AD models (Saito, T., 2014; Olsen, M., 2018; Uruno, A., 2020). Therefore, GFAP was selected as a marker to detect astrocyte activation and astrogliosis in this study, according to previous studies. In previous studies, GFAP is also expressed in the basal state in mature astrocytes in the region stained in this study in adult mice (Tatsumi, K., 2018). We will continue to select appropriate markers and conduct research as you suggested.

Point 3.
When we talk about neuroinflammation, the morphology instead of the number of microglia/ astrocytes may be more informative. 

Response 3.
Thank you for your comment. As you pointed out, we also understand that it is possible to obtain useful information by observing the morphology of glial cells. On the other hand, it is difficult to analyze the morphology of single cells in a in vivo mouse such as the current study. Our method of analyzing glial cells in this study is similar to the method used in previous studies of AD model mice, and we consider that it is a reasonable evaluation method. We are currently conducting AD research using cultured glial cells and would like to apply the morphological analysis as you suggested.

Point 4.
The authors reported that “the protein expression of HIF-1α increased after six months (Figure 6A) (page 10/26, line186-187). However, Fig. 6A shows that there is no difference in the protein expression of HIF-1α between WT and NLPF at 9 and 12 months. Fig. 6B shows the HIF-3α gene expression of WT and NLPF, what about the protein expression of HIF-3α?

Response 4.
Thank you for your comment. The previous expression was confusing, so we have corrected the grammatical expression for tense as follows: Revised words are indicated as "Previous words" to "Revised words" (page number line number). [“after” to “at” (pages 9/22 lines 222).] 
Although we tried to measure the protein expression of HIF-3α as you suggested, we could not have the data about HIF-3α protein expression levels, because of the lack of availability of suitable primary antibody. We searched for a primary antibody to detect HIF-3α in mouse hippocampal tissue, but there were few reliable antibodies with a high number of citations. Even if a suitable one was found (NB100-2529, Novus Biologicals, USA), its production had been discontinued, and antibodies for HIF-3α were also difficult to obtain at present. On the other hand, gene expression of HIF-3α is significantly altered in response to acute hypoxia (Heidbreder, M., 2003). Therefore, we here focused on changes in mRNA expression in HIF-3α. Because protein expression also alters following changes in HIF-3α gene expression (Kumar, H., 2015), basal state protein expression seems to be similarly reduced in the hippocampus at 6 and 9 months of age in NL-P-F mice. HIF-3α protein expression levels are also primarily regulated by changes in HIF-3α gene expression due to transcriptional activation of HIF-1α. Therefore, we regard the examination of gene expression changes as sufficient to evaluate HIF-3α expression levels. However, in future studies, we would like to examine the involvement of HIF-3α in AD by examining the protein expression level of HIF-3α as well.

Point 5.
It seems that the change of both HIF-1α and HIF-3α was not age-dependent in NLPF mice. Why is that? How do you want to put this kind of variation into the context of the pathology and cognitive deficits of the NLPF mice?

Response 5. 
Thank you for your comment. The absence of age-dependent changes in the expression of HIF molecules raises the possibility that HIF expression may differ in different AD pathology stages. For example, it has been reported that microglia increase HIF-1α expression due to phagocytosis of Aβ, but in chronic stages of AD pathology, HIF-1α expression returns to normal levels as a result of the establishment of immune tolerance to Aβ (Baik, S.H., 2019). In our study, mice from six to nine months of age with altered HIF expression begin to show cognitive dysfunction and accumulation of Aβ. These results may indicate that abnormal HIF signaling may be involved in cognitive dysfunction and Aβ accumulation in AD. In the future, we would like to further investigate the molecular mechanism of how toxic conformers play a role in this transient dysregulation of HIF expression.

Point 6. 
The author may want to check the grammar throughout the manuscript including the title. For instance, the text description regarding the mouse ages is confusing.

Response 6.
Thank you for your suggestion. We revised the wording regarding tenses as follows.
1. “after” to “at” (pages 2/22 lines 50)
2. “after” to “at” (pages 2/22 lines 51)
3. “for” to “by” (pages 3/22 lines 93)
4. “after” to “by” (pages 3/22 lines 94)
5. “after” to “at” (pages 3/22 lines 99)
6. “after” to “at” (pages 3/22 lines 103)
7. “after” to “at” (pages 3/22 lines 104)
8. “after” to “by” (pages 3/22 lines 106)
9. “after” to “at” (pages 6/22 lines 156)
10. “for” to “by” (pages 8/22 lines 189)
11. “for” to “by” (pages 8/22 lines 193)
12. “after” to “at” (pages 9/22 lines 222)
13. “after” to “by” (pages 10/22 lines 236)
14. “from” to “by” (pages 10/22 lines 240)
15. “for” to “by” (pages 10/22 lines 241)
16. “after” to “by” (pages 11/22 lines 268)
17. “after” to “by” (pages 11/22 lines 288)
18. “after” to “at” (pages 12/22 lines 309)
19. “after” to “at” (pages 12/22 lines 354)
20. “after” to “at” (pages 13/22 lines 365)
21. “after” to “by” (pages 13/22 lines 375)
22. “after” to “at” (pages 13/22 lines 376)
23. “after” to “by” (pages 18/22 lines 612)
24. “after” to “by” (pages 18/22 lines 618)
25. “after” to “at” (pages 18/22 lines 623)

References
Izuo, N.; Murakami, K.; Fujihara, Y.; Maeda, M.; Saito, T.; Saido, T.C.; Irie, K.; Shimizu, T. An App knock-in mouse inducing the formation of a toxic conformer of Aβ as a model for evaluating only oligomer-induced cognitive decline in Alzheimer's disease. Biochem. Biophys. Res. 2019, 515, 462-467; DOI: 10.1016/j.bbrc.2019.05.131

Saito, T.; Matsuba, Y.; Mihira, N.; Takano, J.; Nilsson, P.; Itohara, S.; Iwata, N.; Saido, T.C. Single App knock-in mouse models of Alzheimer's disease. Nat. Neurosci. 2014, 17, 661-663; DOI: 610.1038/nn.3697; DOI: 10.1038/nn.3697.

Olsen, M.; Aguilar, X.; Sehlin, D.; Fang, X.T.; Antoni, G.; Erlandsson, A.; Syvänen, S. Astroglial responses to amyloid-beta progression in a mouse model of Alzheimer’s disease. Mol. Imaging Biol. 2018, 20, 605-614; DOI: 10.1007/s11307-017-1153-z.

Uruno, A.; Matsumaru, D.; Ryoke, R.; Saito, R.; Kadoguchi, S.; Saigusa, D.; Saito, T.; Saido, T.C.; Kawashima, R.; Yamamoto, M. Nrf2 suppresses oxidative stress and inflammation in App knock-in Alzheimer’s disease model mice. Mol. Cell. Biol. 2020, 40, e00467-00419; DOI: 10.1128/MCB.00467-19.

Tatsumi, K.; Isonishi, A.; Yamasaki, M.; Kawabe, Y.; Morita-Takemura, S.; Nakahara, K.; Terada, Y.; Shinjo, T.; Okuda, H.; Tanaka, T. Olig2-lineage astrocytes: a distinct subtype of astrocytes that differs from GFAP astrocytes. Front. Neuroanat. 2018, 12, 8; DOI: 10.3389/fnana.2018.00008.
Heidbreder, M.; Fröhlich, F.; Jöhren, O.; Dendorfer, A.; Qadri, F.; Dominiak, P. Hypoxia rapidly activates HIF‐3α mRNA expression. FASEB J. 2003, 17, 1-19; DOI: 10.1096/fj.02-0963fje.

Kumar, H.; Lim, J.-H.; Kim, I.-S.; Choi, D.-K. Differential regulation of HIF-3α in LPS-induced BV-2 microglial cells: Comparison and characterization with HIF-1α. Brain Res. 2015, 1610, 33-41;DOI: 10.1016/j.brainres.2015.03.046.

Baik, S.H.; Kang, S.; Lee, W.; Choi, H.; Chung, S.; Kim, J.-I.; Mook-Jung, I. A breakdown in metabolic reprogramming causes microglia dysfunction in Alzheimer's disease. Cell Metab. 2019, 30, 493-507. e496; DOI: 10.1016/j.cmet.2019.06.005.

Round 2

Reviewer 3 Report

My questions have been addressed in the revised manuscript. I have no further questions.